# Flame-Retarded Rigid Polyurethane Foam Composites with the Incorporation of Steel Slag/Dimelamine Pyrophosphate System: A New Strategy for Utilizing Metallurgical Solid Waste

**DOI:** 10.3390/molecules27248892

**Published:** 2022-12-14

**Authors:** Mingxin Zhu, Sujie Yang, Zhiying Liu, Shunlong Pan, Xiuyu Liu

**Affiliations:** 1College of Environmental Science and Engineering, Nanjing Tech University, Nanjing 211816, China; 2School of Architecture and Civil Engineering, Anhui University of Technology, Ma’anshan 243002, China

**Keywords:** rigid polyurethane, steel slag, dimelamine pyrophosphate, solid waste utilization, flame retardant

## Abstract

Rigid polyurethane (RPUF) was widely used in external wall insulation materials due to its good thermal insulation performance. In this study, a series of RPUF and RPUF-R composites were prepared using steel slag (SS) and dimelamine pyrophosphate (DMPY) as flame retardants. The RPUF composites were characterized by thermogravimetric (TG), limiting oxygen index (LOI), cone calorimetry (CCT), and thermogravimetric infrared coupling (TG-FTIR). The results showed that the LOI of the RPUF-R composites with DMPY/SS loading all reached the combustible material level (22.0 vol%~27.0 vol%) and passed UL-94 V0. RPUF-3 with DMPY/SS system loading exhibited the lowest pHRR and THR values of 134.9 kW/m^2^ and 16.16 MJ/m^2^, which were 54.5% and 42.7% lower than those of unmodified RPUF, respectively. Additionally, PO· and PO_2_· free radicals produced by pyrolysis of DMPY could capture high energy free radicals, such as H·, O·, and OH·, produced by degradation of RPUF matrix, effectively blocking the free radical chain reaction of composite materials. The metal oxides in SS reacted with the polymetaphosphoric acid produced by the pyrolysis of DMPY in combustion. It covered the surface of the carbon layer, significantly insulating heat and mass transport in the combustion area, endowing RPUF composites with excellent fire performance. This work not only provides a novel strategy for the fabrication of high-performance RPUF composites, but also elucidates a method of utilizing metallurgical solid waste.

## 1. Introduction

Rigid polyurethane (RPUF) is widely used in thermal insulation, refrigeration, architectural decoration, amongst others, due to its excellent thermal insulation, mechanical properties, and environmental resistance [1,2,3,4,5]. However, due to the porous structure and organic skeleton of RPUF, it is also easily ignited and releases toxic gases, such as HCN, CO, etc. The limiting oxygen index (LOI) of RPUF is only approximately 19.0 vol%, with intense dripping in fire [6]. The above disadvantages significantly limit the further application of RPUF. Thus, how to reduce the toxic gases generated during the combustion process of RPUF and enhance its fire performance has attracted much scholarly attention [7].

Steelmaking as a basic industry and steel products have penetrated all areas of life. According to statistics, the global crude steel production in 2020 was as high as 1.878 billion tons [8]. Steel slag (SS), as an unavoidable solid waste in the steelmaking process, is partially stored in piles, causing excessive resource waste and environmental pollution. Thus, exploring the potential value of SS and improving its comprehensive utilization rate is the current focus of researchers in the solid waste industry. Additionally, exploring the high value-added utilization pathway of steel slag is an important step to accelerating the improvement of the green low-carbon circular economy system. In recent years, many studies on the utilization of SS have been conducted, for example, in cement fields, mine backfilling, and soil remediation, amongst others [9,10,11,12].

Steel slag (SS) is rich in metal oxides such as CaO, SiO_2_, and Al_2_O_3_. These compounds exhibit excellent catalytic carbonization and smoke suppression effects in combustion and thus have potential applications in fire-retarding fields [13,14]. Xu et al. [15] prepared melamine-cyanuric acid fumed silica (MCA-SiO_2_) by reacting melamine (ME) and cyanuric acid (CA) in aqueous suspension and deposited it on fumed silica. Subsequently, GF-PP and IFR containing MCA-SiO_2_ were introduced to prepare GF-PP/IFR-(MCA-SiO_2_) composites via a melt blending technology. The results showed that the heat release rate (HRR) of GF-PP/IFR-MCA composites decreased, and the flame-retardant property increased with the addition of SiO_2_. When the content of SiO_2_ was 20 wt%, the flame-retardant property of the composites reached a UL-94 V-0 rating, and the limiting oxygen index (LOI) increased to 32.4 vol%. Tang et al. [16] modified steel slag (SS) by DOPO-derived silanes through a sol–gel reaction, synergizing the modified steel slag (mSS) with expanded graphite (EG) in rigid polyurethane foam. The results showed that when the mSS/EG additions reached 20 wt%, the RPUF composite exhibited the best overall performance. The peak heat release rate (pHRR) and total heat release (THR) of the composite were decreased by 55% and 47%, respectively, and ultimate oxygen index increased to 24.0 vol% and UL-94 V0 rating in flame-retardant tests.

Dimelamine pyrophosphate (DMPY) is an environmentally friendly non-halogen phosphorus–nitrogen flame retardant commonly used in polyolefins, coatings, and fibers [17]. DMPY molecules contain triazine rings, which have excellent charring ability [18,19,20]. Thus, DMPY has attracted much scholarly attention and use alone or as a compound with other flame retardants. Liu et al. [21] synthesized DMPY using sodium pyrophosphate and melamine as raw materials, which was further introduced into thermoplastic epoxy resin (EP) composites to enhance its fire performance. It was observed that the flame retardancy and mechanical properties of EP/DMPY composites were significantly improved compared to pure EP, and the ultimate oxygen index of the composites was increased to 28.7 vol% when 9 wt% DMPY was incorporated. Liu et al. [22] combined DMPY and aluminum diethylphosphinate (ADP) as a synergistic system (FRs), which was further used in UPR thermoset composites. The results showed that the decomposition of FRs produced pyrophosphate, polyphosphate, and metaphosphate, which catalyzed the degradation of the UPR matrix to form coherent, partially graphitized, dense carbon layers.

Currently, the direct addition of flame retardants is the simplest strategy to improve the flame-retardant properties of polyurethane [23,24]. However, there are no reports on the applications of DMPY in RPUF. Thus, in this study, a series of RPUF and RPUF-R composites were prepared with DMPY and SS as flame retardants, and the flame-retardant properties, combustion properties, and morphological changes of the composites were investigated using limiting oxygen index (LOI), vertical combustion (UL-94), cone calorimetry (CCT), and scanning electron microscopy (SEM) test methods. This work provides a novel strategy for the high value-added utilization of SS and fabricating high-performance RPUF composites.

## 2. Results and Discussion

### 2.1. Bubble Structure of RPUF and RPUF-R Composites

SEM was a useful tool for investigation particle size and micromorphology of materials [25,26]. The morphology of RPUF and RPUF-R composites were analyzed by SEM. As observed from Figure 1a, the pure RPUF had a bubble structure with uniform pores structure, which was consistent with a previous report [27]. Additionally, as shown in red circles in Figure 1a, some big bubble structures were observed, which may come from the uneven distribution of water (blowing agent) in the foaming process. As presented in Figure 1a1, pure RPUF was endowed with a thin wall for bubble structure. As can been found in Figure 1b, the big bubble structures were also observed when 20 wt% DMPY was added. Figure 1b1 confirmed that the wall of blister in RPUF-1 thickened and the density increased significantly. Meanwhile, it also could be found that DMPY particles remained on the surface of the composite and some of the blister holes were broken (as the yellows arrow point), which may be due to the excessive amount of DMPY increasing viscosity and promoting the agglomeration of flame-retardant particles. When SS replaced part of DMPY, it could be observed that the agglomeration of the RPUF-3 composite was reduced, but the hole wall was still partially damaged (points by red circle), indicating that the addition of inorganic SS was conducive to improving the compatibility of DMPY and RPUF matrix. When SS was added to the RPUF matrix completely instead of DMPY, it could be observed from Figure 1d1 that the fracture surface of the vesicle pore wall of the RPUF-1 composite was smooth and the wall layer was thickened (points by red circle), which indicated that the inorganic SS particles possessed excellent compatibility with the RPUF matrix. It can be found from Figure 1b–d that when SS, DMPY, and their combination were introduced into RPUF, the pore size of the composites were smaller than that of unmodified RPUF, which was mainly due to the nucleation effect of powdery flame retardant.

### 2.2. Physical Properties of RPUF and RPUF-R Composites

Apparent density was a very important physical property index for composites [28,29]. As shown in Table 1, pure RPUF exhibited density of 45.28 kg/m^3^, with thermal conductivity of 0.0373 W/m·k, and compressive strength of 0.325 KPa. When 20 wt% DMPY was added, the apparent density of RPUF-1 was increased to 71.84 kg/m^3^, which was significantly higher than that of unmodified RPUF. This may be due to the fact that the density of DMPY was much higher than that of the RPUF matrix. Additionally, the apparent density of RPUF-1 was increased, which may come from the enhancement of nucleation ability and the decrease in cell size after DMPY loading [30]. With the increase in the SS and decrease in DMPY, the apparent density of the RPUF-R composites were gradually decreased. This may come from the fact that SS contained free f-CaO and f-MgO components. When SS met water, it underwent hydration reaction and generated Ca(OH)_2_ and Mg(OH)_2_, resulting in volume expansion. The total mass of the composite was a constant, with the increase of SS addition, the expansion rate of the composites gradually increased, thus resulting in decrease in the apparent density for the composites [31]. It was observed that the thermal conductivity of RPUF-R composites was higher than that of unmodified RPUF. This was mainly ascribed to the damage of part of the cell structure caused by the addition of powdery flame-retardant particles. Compression strength was another important index to characterize performance of rigid polyurethane foam [32]. In general, the compressive strength of polyurethane rigid foam increased with the increase in density. RPUF-R composites exhibited lower compressive strength than that of pure RPUF, which was related to the excessive addition of powdery flame retardants and the destruction of hydrogen bonds in RPUF composites.

### 2.3. Flame Retardant Properties of RPUF and RPUF-R Composites

LOI and UL-94 were important methods for evaluating the flame retardancy of polymer materials [33]. As shown in Table 2, the LOI value of pure RPUF was only 19.0 vol%, which was a flammable material and exhibited molten dripping in combustion. When 20 wt% DMPY was added, the LOI value of RPUF-1 increased to 24.8 vol% with UL-94 V-0 rating, and the dripping phenomenon disappeared. This was due to the existence of DMPY, which produced polymetaphosphate in the decomposition process, and promoted the carbonization of the RPUF matrix. At the same time, polymetaphosphate was a viscous glass material. After dehydration and carbonization, the glass material could attach to the surface of the carbon layer to form a liquid film and achieve the purpose of flame retardancy [34]. With the replacement of DMPY by SS, the LOI values of the composites were in the range of 22.0 vol%~25 vol%, which reached the combustible material rating (22.0 vol%~27.0 vol%) compared with flammable material (<22.0 vol%) of pure RPUF. Additionally, the RPUF composites with DMPY and SS loading all could pass UL-94 V0 flame retardant rating [35]. This may be due to the fact that SS contained Al_2_O_3_, MgO, and CaO, which combined with DMPY to form a protective layer and prevent the further combustion of the underlying composites. When 20 wt% of SS was added to completely replace DMPY, the LOI value of RPUF-5 composite was 20.4 vol%, which was 1.4 vol% higher compared to unmodified RPUF and failed to pass UL-94 V-0 flame retardant rating. It was also found that the melt dripping phenomenon disappeared. The above results indicated that the addition of SS alone has some effect on RPUF flame retardancy, but the combination of SS with DMPY can be effective for flame retardancy enhancement of RPUF-R composites.

### 2.4. Thermal Stability of RPUF and RPUF-R Composites

Thermogravimetry was a useful tool for investigating the thermal stability of polymer composites [36]. Herein, the thermal stability of RPUF and RPUF-R composites was characterized by thermogravimetry (TGA). Figure 2 showed the thermogravimetric (TG) and thermal weight loss (DTG) curves of RPUF and RPUF-R composites, and the related data were listed in Table 3. As shown in the table, the initial degradation temperature (T_-5wt%_) of unmodified RPUF was 271 °C. The T_-5wt%_ of RPUF-1, RPUF-3, and RPUF-5 composites were higher than that of pure RPUF, indicating that the addition of DMPY and SS inhibited the initial decomposition of RPUF composites. The whole pyrolysis process of the composites was divided into two stages. The first stage in the interval of 300–350 °C corresponded to the degradation of the hard segments of the polyurethane molecular chain, and the second stage in the interval of 450–500 °C could be ascribed to the degradation of the soft segments in the polyurethane molecular chain, accompanied by the generation of CO [37,38]. It can also be noted from Table 4 that the maximum degradation temperature (T_max2_) of RPUF-1, RPUF-3, and RPUF-5 in the second stage were higher than that of pure RPUF, which may be due to the fact that the polymetaphosphate generated by pyrolysis of DMPY promoted the dehydration and carbonization of RPUF composites, and free PO· radicals captured high-energy radicals such as H·, O·, and OH·, which could effectively block the free radical chain reaction in combustion [39]. At the same time, the metal oxides in SS chelated with the polyphosphoric acid generated by DMPY pyrolysis to form metal salts. It could be retained on the surface of the carbon layer, which was a benefit for isolating oxygen and heat, preventing the further combustion of the composites and improving the thermal stability of the composites [40]. The residual carbon of RPUF-1, RPUF-3, and RPUF-5 composites at 700 °C were higher than those of pure RPUF. All these results indicated that the addition of DMPY and SS could improve the high temperature thermal stability of RPUF-R composites.

### 2.5. Combustion Properties of RPUF and RPUF-R Composites

Cone calorimeter was widely used in evaluating the combustion performance of materials [41]. Figure 3 and Figure 4 showed the conical calorimetric data plots of RPUF and RPUF-R composites, and the related data were listed in Table 4. The pure RPUF reached the maximum heat release rate (pHRR) at 72 s, with the highest value of 296.5 kW/m^2^. The pHRR values of RPUF-1, RPUF-3, and RPUF-5 composites were 145.1 kW/m^2^, 134.9 kW/m^2^, and 188.6 kW/m^2^, respectively, which were 51.1%, 54.5%, and 36.4% lower than those of pure RPUF. The HRR values of RPUF-R composites decreased rapidly after reaching pHRR, which were lower than that of RPUF. RPUF-3 was endowed with the lowest pHRR value, suggesting the synergistic effect between SS and DMPY in RPUF composites. It can be seen from Figure 3b that the THR value of pure RPUF was 28.18 MJ/m^2^, and THR values of RPUF-R were lower than that of pure RPUF. Among them, the decrease in the RPUF-3 composite was the most obvious, with THR value of 16.16 MJ/m^2^, which was 42.7% lower than that of pure RPUF, implying that DMPY/SS could synergistically reduce the heat release of RPUF composite in combustion. This result was highly consistent with the flame retardant test.

Furthermore, typical gaseous products and mass vs. time curves were given in Figure 4. CO was a deadly toxic gas produced in a fire accident [42]. As observed in Figure 4, the CO production of RPUF-R composites showed a decreasing trend, which was mainly due to the possible adsorption of CO by the metal ions contained in SS. Additionally, part of the CO could be oxidized into CO_2_ by oxidation reaction under high temperature conditions [43]. Fire Performance Index (FPI) and Fire Spread Index (FGI) were often used to evaluate the fire safety of flame-retardant polymer composites, where FPI was the ratio of PHRR to peak burning time of the composite materials (T_P_), and FGI was the ratio of TTI to PHRR. The higher the FPI value, the smaller the FGI value, and the smaller the fire hazard [44,45]. As shown in Table 5, the FPI values of RPUF, RPUF-1, RPUF-3, and RPUF-5 composites were 0.0169 m^2^·s/kW, 0.0207 m^2^·s/kW, 0.0297 m^2^·s/kW, and 0.0159 m^2^·s/kW, respectively, and the FGI values were 4.12 kW/m^2^·s, 2.13 kW/m^2^·s, and 1.87 kW/m^2^·s, and 2.95 kW/m^2^·s, respectively. Compared with pure RPUF, the fire hazards of RPUF-R composites were all significantly reduced, in which RPUF-3 exhibited the most significant effect, indicating that the addition of DMPY/SS synergistically reduced the fire safety of RPUF composites.

### 2.6. Analysis of Gas Phase Products of RPUF and RPUF-R Composites

Thermogravimetric infrared coupling (TG-FTIR) was a common method to analyze the gas phase products and work mechanism of flame-retarding composites [46]. Figure 5 provided TG-FTIR 3D Spectra of RPUF and RPUF-R Composites. It can be observed from the figure that the gas phase products of RPUF and RPUF-R composites were mainly distributed in the bands of 3500–4000 cm^−1^, 2500–3400 cm^−1^, 2100–2400 cm^−1^, 1500–2000 cm^−1^, 700–1300 cm^−1^. Figure 6 showed the FTIR spectra of the gas products of RPUF and RPUF-R composites at different times. The characteristic peaks at around 3818 cm^−1^ and 2930 cm^−1^ corresponded to the stretching vibration of the N–H bond and the stretching vibration of the C–H bond in hydrocarbons, respectively [47]. The characteristic peaks of CO_2_ and -NCO were 2384 cm^−1^ and 2306 cm^−1^, respectively. The characteristic peaks at around 1604 cm^−1^ and 1517 cm^−1^ were attributed to aromatic compounds and esters [48]. The characteristic peak of HCN was found at 721 cm^−1^, which was consistent with a previous report. From Figure 6a, it could be found that the release intensity of CO_2_ and -NCO from the RPUF-3 composite was significantly reduced compared to the other three composites. Additionally, the release intensity of HCN was reduced compared to pure RPUF, indicating that the chelation reaction between DMPY and SS inhibited the release of toxic and hazardous gases associated with the decomposition of RPUF matrix. Figure 6b–e showed the variation curves of typical gaseous products intensity for RPUF and RPUF-R composites with time. As observed from Figure 6b, the CO_2_ release of pure RPUF started at around 12 min, while the CO_2_ generation time of the other three composites was delayed, implying that the addition of DMPY and SS inhibited the initial degradation of the composites and improved their thermal stability, which was consistent with the thermogravimetric test.

### 2.7. Carbon Slag Analysis of RPUF and RPUF-R Composites

Carbon residue in RPUF and RPUF-R composites was obtained after calcination in muffle furnace, which were further investigated by SEM [49]. As shown in Figure 7a, the carbon layer of pure RPUF exhibited a thin and smooth structure after calcination (as shown by red circle), and the char layer was lax, which could not inhibit the mass and heat transport in combustion. As shown in Figure 7b, when 20 wt% DMPY was added, the carbon layer compactness of RPUF-1 composite was significantly improved compared with that of unmodified RPUF, but some pore structures (as observed from the red circle) was observed, which may be due to the poor compatibility between DMPY and RPUF matrix and the uneven dispersion caused by excessive addition of DMPY. When 20 wt% of SS was added, the compactness of carbon layer for RPUF-5 was also improved. It was also found that the surface of the carbon layer was loaded with a large amount of metal oxides and SS particles (pointed by yellow arrow), which acted as a barrier to prevent the transport of heat and mass. When 20 wt% DMPY/SS (1:1) was added, many obvious worm structures and dense carbon layers were observed in RPUF-3 (as shown by red circle). This was mainly due to the combination of the catalytic carbonization effect of polyphosphate resulting from DMPY decomposition, and the crosslinking effect of metal ions that came from SS during combustion. Based on the above investigation, we came to the conclusion that the synergistic effect of DMPY/SS system effectively inhibited the heat and mass transfer in combustion region, endowing RPUF-3 with excellent fire-retarding performance.

Raman spectroscopy is a common method to measure the graphitization degree of carbon materials [50]. Figure 8 shows the Raman spectra of RPUF and RPUF-R composites. As shown in the figure, two broadband bands were present in all the samples. The D band at around 1360 cm^−1^ corresponded to the amorphous phase consisting of disordered carbon atoms, and the G band at 1590 cm^−1^ was attributed to crystalline phase consisting of graphited carbon atoms [51]. The area ratio of I_D_/I_G_ was usually used to represent the graphitization degree of the sample. The smaller the ratio of I_D_/I_G_, the higher graphitization degree of carbon residue with better fire resistance [52,53]. As observed from Figure 8, the I_D_/I_G_ values of the carbon residues for RPUF-R composites were all lower than that of unmodified RPUF, suggesting that the usage, either alone or combination, of SS and DMPY could endow RPUF-R composites with enhanced fire-retarding performance.

## 3. Experiment

### 3.1. Experimental Materials

Polyether polyols (LY-4110), triethylenediamine solution (A33), industrial grade, were all provided by Jiangsu Lvyuan New Materials Technology Co., Ltd. (Nantong, China). Trimethylene polyphenyl polyisocyanate (PM-200, industrial grade) was purchased from Wanhua Chemical Group Co., Ltd. (Yantai, China). Silicone oil foam stabilizer (AK-8805, industrial grade) was purchased from Jining Hengtai Chemical Co., Ltd. (Jining, China). Triethanolamine (TEOA, Chemically Pure), was purchased from Chemical Reagent Co., Ltd. (Shanghai, China). Dibutyltin dilaurate (LC, chemical purity) was purchased from Air Chemical Co., Ltd. (Delaware, Ameica). Dimelamine pyrophosphate (DMPY) was purchased from Dongguan Shengde New Materials Co., Ltd. (Dongguan, China). Steel slag (SS) was kindly provided by Masteel Group. None of the above chemicals have been further purified.

### 3.2. Sample Foaming

A series of RPUF and RPUF-R composites were prepared by one-step aqueous foaming process. The composition ratio of the composites was shown in Table 5. Firstly, LY-4110, LC, A33, AK-8005, TEA, SS, DMPY, and distilled water were added into a 500 mL beaker and stirred at 2000 rpm for 1 min. Then, the weighed PM-200 was further added into the beaker with intense stirring for another 10 s until the mixture turned white, which was then poured into the mold quickly. The samples were further reacted at 80 °C for 6 h to complete the polymerization process. Then, the obtained sample was cut into suitable sizes for the following testing.

### 3.3. Sample Characterization

Limit oxygen index (LOI): JF-3 oxygen index analyzer (Nanjing Jiangning District Instrument Analysis Factory) was applied to test the limit oxygen index of the composites with sample size of 127 mm × 10 mm × 10 mm. Vertical combustion (UL-94): Horizontal vertical combustion tester (CZF-2) was introduced to test combustion rating of RPUF and RPUF composites with sample size of 127 mm × 13 mm × 10 mm. Mechanical properties: The DCS-5000 universal material testing machine was applied to test the compressive strength of the RPUF and RPUF-R composites with sample size of 50 mm × 50 mm × 40 mm, and each sample was tested for five times to obtain average value. Thermal conductivity test: TC3000E thermal conductivity meter was used to test the thermal conductivity of the composites with sample size of 50 mm × 50 mm × 25 mm, and the average value was obtained by test for three measurements. Cone calorimeter (CCT): 6180 (Siemens analyzer) cone calorimetry was used to test the heat release rate and smoke release of the composites, where the radiation flux was 35 kW/m^2^ and the sample size was 100 mm × 100 mm × 25 mm. Scanning electron microscopy (SEM): The morphology of the composites and carbon slags were investigated by JSM-6490LV scanning electron microscope. In order to enhance the electrical conductivity of the samples, the samples were sprayed with conductive layer. Laser confocal Raman spectroscopy (Roman): The carbon slag was obtained by calcining the composites in a muffle furnace at 600 °C for 10 min, and then tested by laser confocal Raman spectroscopy (LabRAM HR Evolution, HORIBA Scientific) to investigate the graphitization degree of the carbon slag for the composites.

## 4. Conclusions

In this work, a series of RPUF and RPUF-R composites were fabricated by one-step all-water foaming method using DMPY and SS as flame retardants. Additionally, the effects of DMPY and SS on the microscopic morphology, thermal stability, flame retardancy, combustion properties, gas phase products, and carbon residue of the composites were systematically investigated. The LOI test confirmed that RPUF/DMPY and RPUF/DMPY/SS composites all passed UL 94 V0 rating, and the LOI values reached the combustible material grade (22.0 vol%~27.0 vol%). The RPUF/SS composite with SS alone cannot pass UL 94 V0 rating with relatively low LOI value of 20.4 vol%. RPUF-R composites were observed dripping phenomenon disappeared, implying the enhanced fire hazard of the composites. TG test showed that the residual carbon value and T_-5wt%_ of RPUF-1, RPUF-3, and RPUF-5 composites at 700 °C were higher than those of unmodified RPUF, confirming that the addition of DMPY and SS could significantly improve the high temperature thermal stability of the composites. Cone colorimetry test showed that RPUF-3 composite possessed the lowest pHRR and THR values of 134.9 kW/m^2^ and 16.16 MJ/m^2^, indicating that the combination of DMPY and SS could inhibit the heat release of the composites in combustion. The carbon slag investigation showed that the DMPY/SS synergistic system improved the denseness and graphitization degree of the carbon layer for RPU composites, thus inhibiting the heat and mass transport of the composites in combustion. The above results show that DMPY/SS can be a feasible strategy for fabricating fire-retardant RPUF composites, which also paves a new way for solid waste utilization.

## Figures and Tables

**Figure 1 molecules-27-08892-f001:**
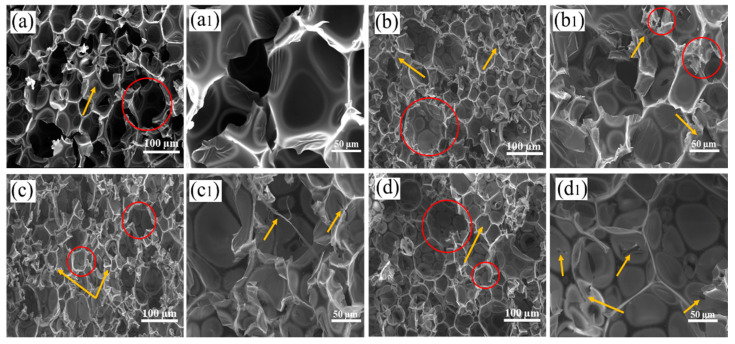
SEM images of RPUF and RPUF-R composites: (**a**,**a1**) RPUF; (**b**,**b1**) RPUF-1; (**c**,**c1**) RPUF-3; (**d**,**d1**) RPUF-5.

**Figure 2 molecules-27-08892-f002:**
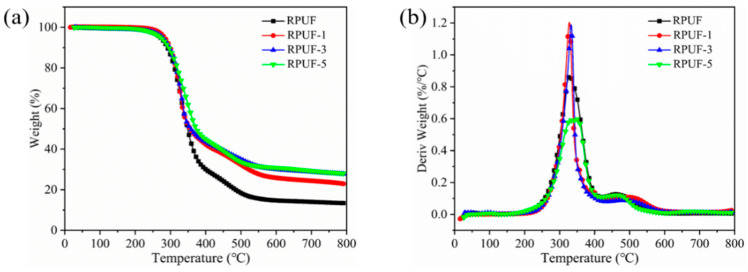
(**a**) thermogravimetric (TG) and (**b**) thermal weight loss (DTG) curves of RPUF and RPUF-R composites.

**Figure 3 molecules-27-08892-f003:**
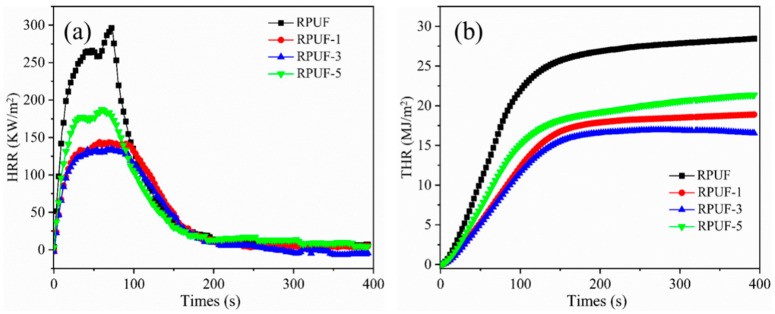
(**a**) HRR and (**b**) THRcurves of RPUF and RPUF-R composites.

**Figure 4 molecules-27-08892-f004:**
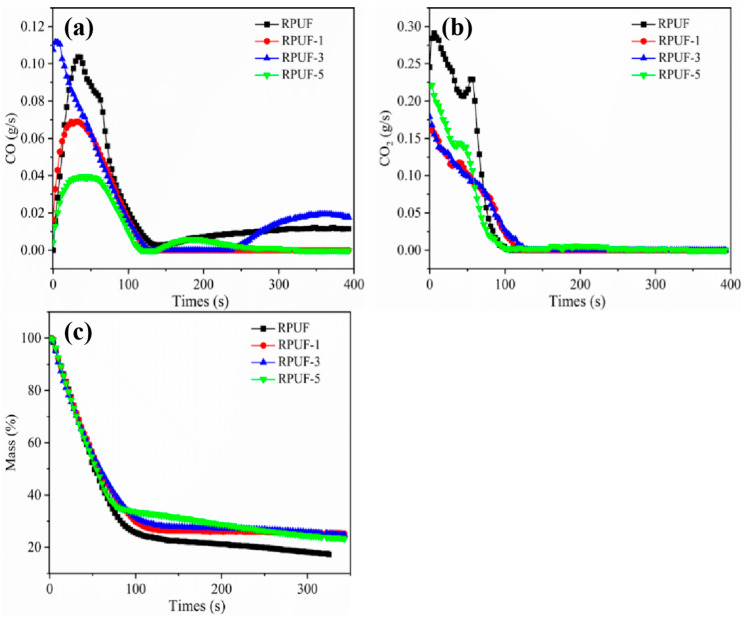
(**a**) CO, (**b**) CO_2_, and (**c**) Mass curves of RPUF and RPUF-R composites.

**Figure 5 molecules-27-08892-f005:**
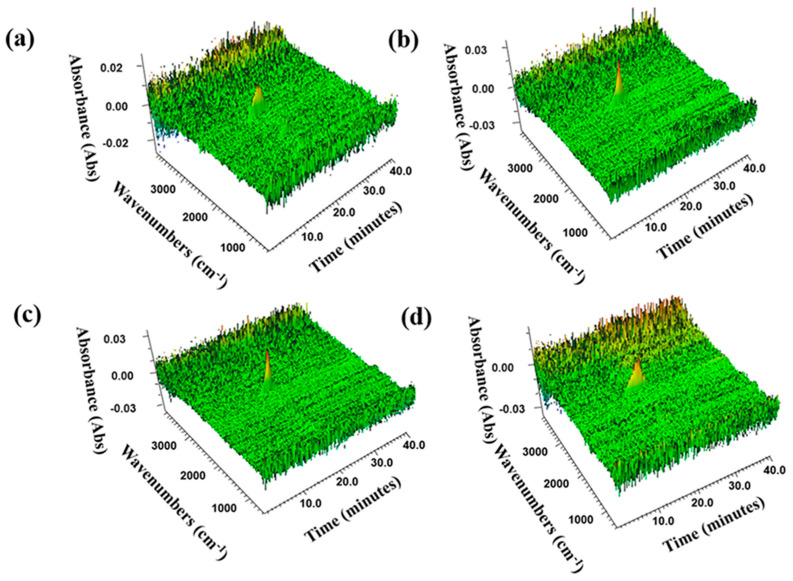
TG-FTIR 3D images of RPUF and RPUF-R composites: (**a**) RPUF; (**b**) RPUF-1; (**c**) RPUF-3; (**d**) RPUF-5.

**Figure 6 molecules-27-08892-f006:**
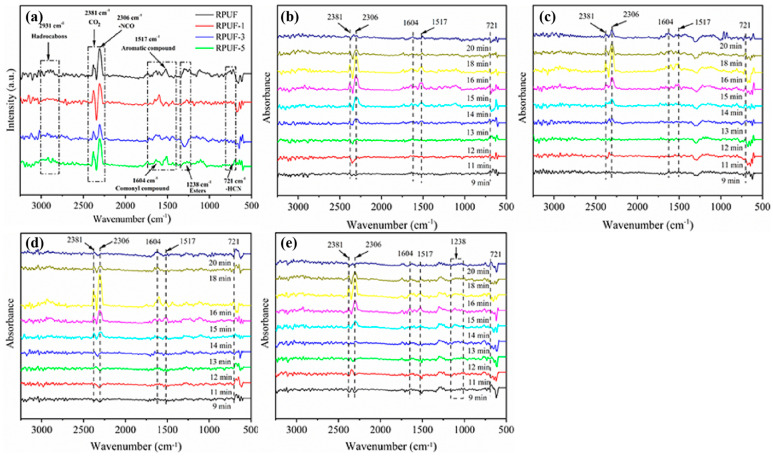
FTIR spectra of thermal decomposition products of RPUF and RPUF-R composites: (**a**) FTIR spectra at T_max_; (**b**) RPUF; (**c**) RPUF-1; (**d**) RPUF-3; (**e**) RPUF-5.

**Figure 7 molecules-27-08892-f007:**
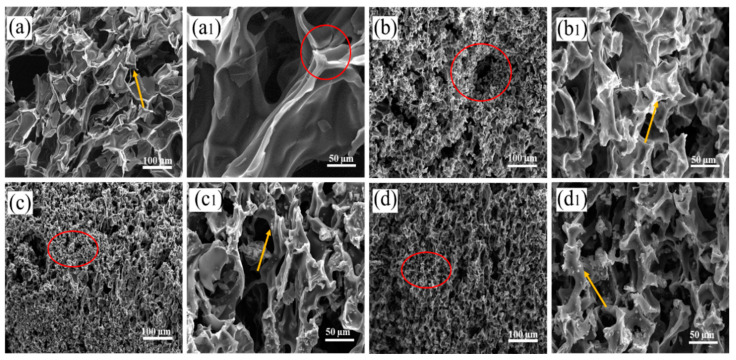
SEM images of RPUF and RPUF-R composite carbon slag: (**a**,**a1**) RPUF; (**b**,**b1**) RPUF-1; (**c**,**c1**) RPUF-3; (**d**,**d1**) RPUF-5.

**Figure 8 molecules-27-08892-f008:**
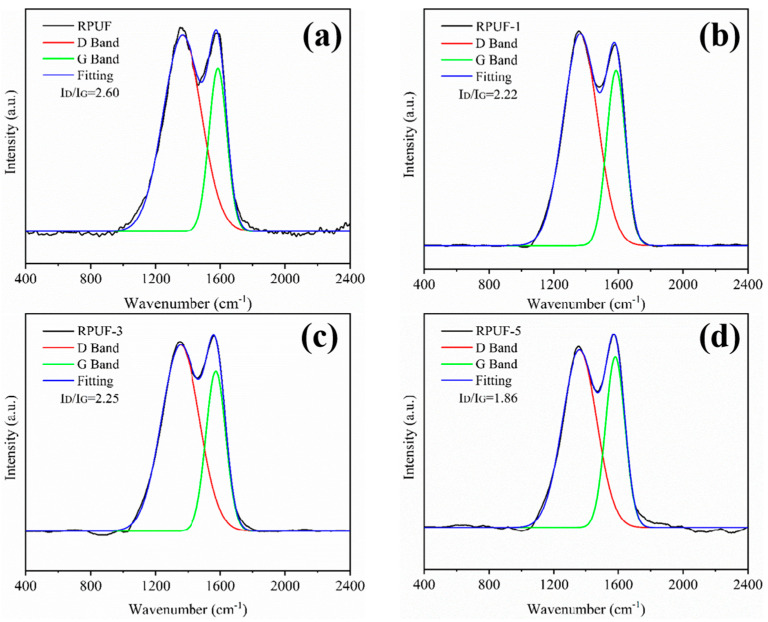
Raman spectra of RPUF and RPUF-R composites: (**a**) RPUF; (**b**) RPUF-1; (**c**) RPUF-3; (**d**) RPUF-5.

**Table 1 molecules-27-08892-t001:** Apparent density, thermal conductivity, and compressive strength of RPUF and RPUF-R composites.

Sample	Thermal Conductivity/(W/m·k)	Compressive Strength/(MPa)	Apparent Density/(kg/m^3^)
RPUF	0.0373	0.325	45.28
RPUF-1	0.0392	0.315	71.84
RPUF-2	0.0380	0.289	63.80
RPUF-3	0.0389	0.273	61.44
RPUF-4	0.0381	0.257	61.20
RPUF-5	0.0369	0.240	59.25

**Table 2 molecules-27-08892-t002:** LOI, UL-94 data for RPUF and RPUF-R composites.

Sample	LOI/vol%	UL-94. 3.2 mm Bar		
		t_1_/t_2_^a^ (s)	Dripping	Rating
RPUF	19.0	BC^b^	Y	NR^c^
RPUF-1	24.8	4.37/0	N	V-0
RPUF-2	23.7	6.42/0	N	V-0
RPUF-3	23.4	6.84/0	N	V-0
RPUF-4	22.4	8.72/0	N	V-0
RPUF-5	20.4	BC	N	NR

Note: a t_1_, t_2_—the first, second ignition after the burning time; b BC—burn to fixture; c NR—no rating.

**Table 3 molecules-27-08892-t003:** Thermogravimetric data of RPUF and RPUF-R composites (nitrogen atmosphere).

Sample	T_−5wt%_/°C	T_max1_/°C	T_max2_/°C	700 °C Carbon Residue/wt%
RPUF	271	330	460	14.0
RPUF-1	283	328	490	24.4
RPUF-3	277	333	470	28.8
RPUF-5	272	350	463	29.3

**Table 4 molecules-27-08892-t004:** Cone calorimetric data of RPUF, RPUF-R composites.

Sample	RPUF	RPUF-1	RPUF-3	RPUF-5
TTI (s)	5	3	4	3
Tp (s)	72	68	72	64
Td (s)	390	415	516	441
pHRR (kW/m^2^)	296.5	145.1	134.9	188.6
THR (MJ/m^2^)	28.18	18.95	16.16	21.52
FPI (m^2^·s/kW)	0.0169	0.0207	0.0297	0.0159
FGI (kW/m^2^·s)	4.12	2.13	1.87	2.95
CY (wt%)	17.27	25.22	24.77	23.29

Note: TTI, time to ignition; Tp, time to pHRR; Td, continuous combustion time; pHRR, peak of release rate; THR, total heat release; FPI, fire performance index; FGI, fire growth rate index; CY, char yield.

**Table 5 molecules-27-08892-t005:** Composition of RPUF and RPUF-R composites.

Sample	LY-4110 /g	PM-200 /g	LC /g	AK-8805 /g	A33 /g	TEOA /g	Water /g	SS /g	DMPY /g
RPUF	100	135	0.5	2	1	3	2	0	0
RPUF-1	100	135	0.5	2	1	3	2	60.9	0
RPUF-2	100	135	0.5	2	1	3	2	40.6	20.3
RPUF-3	100	135	0.5	2	1	3	2	30.45	30.45
RPUF-4	100	135	0.5	2	1	3	2	20.3	40.6
RPUF-5	100	135	0.5	2	1	3	2	0	60.9

Note: LY-4110, polyether polyol; PM-200, polyaryl polymethylene isocyanate; LC, dibutyltin dilaurate, AK-8805, silicone surfactant; A33, triethylenediamine; TEOA, triethanolamine, Water, blowing agent; SS, steel slag; DMPY, dimelamine pyrophosphate.

## Data Availability

The data will be available on request.

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
