# Peer review of "Flame-Retarded Rigid Polyurethane Foam Composites with the Incorporation of Steel Slag/Dimelamine Pyrophosphate System: A New Strategy for Utilizing Metallurgical Solid Waste"

_molecules, 2022, doi:10.3390/molecules27248892_

Round 1

Reviewer 1 Report

The authors have used steel slag (SS) and dimelamine pyrophosphate (DMPY) to reduce the flammability of rigid polyurethane foam (RPUF). Foam with and without added flame retardant (FR) is well characterized and there is clearly a significant improvement. Although the synergy between SS and DMPY is modest, the overall study demonstrates a potential use for a major waste product, which lends merit to this study. There are a number of issues that should be addressed prior to publication of this work:

1. The authors use past tense throughout the manuscript when they should be using present tense. Whenever possible (and grammatically correct), present tense should be used to make the text more engaging. The first sentence of the abstract incorrectly uses past tense and there are many more (most) sentences throughout the manuscript that do this.

2. In the second paragraph of the Introduction (on page 1), the sentence that begins with "Steel slag (SS) is rich..." should become the first sentence of this paragraph because it defines "SS". In the current organization, the abbreviation SS is used without being defined.

3. Section 2.3 can be one single paragraph.

4. What is the average particle size of the SS used here? This should be measured and reported in the manuscript.

5. The caption of Figure 1 should include a description of what the red circles and yellow arrows indicate.

6. The references are not correct throughout the manuscript text. They appear as [0] in many locations.

7. The first sentence of Section 3.7 is not actually a sentence (i.e., it is an incomplete sentence). The entire manuscript text needs copyediting to fix clumsy English grammar.

8. In the Conclusions (on page 11), line 327 does not need to list the tools used for characterization. Deleting this list from the sentence will make it more concise and impactful. 

Author Response

Journal: Molecules

Manuscript Number: molecules-2017594

Title: Flame retarded rigid polyurethane foam composites with incorporation of steel slag/dimelamine pyrophosphate system: a new strategy for utilizing of metallurgical solid waste

Author(s): Mingxin Zhu, Sujie Yang, Zhiying Liu, Shunlong Pan, Xiuyu Liu

Thanks a lot for your attention and the reviewers’ evaluation and comments on our manuscript (molecules-2017594). We have modified the manuscript according to your helpful advice and reviewers’ detailed suggestions with the correction sections. At the same time, the grammatical errors of this manuscript have been carefully revised. Please find the point-by-point response to the reviewers.

Yours sincerely,

Mingxin Zhu

Reviewer 1:

The authors have used steel slag (SS) and dimelamine pyrophosphate (DMPY) to reduce the flammability of rigid polyurethane foam (RPUF). Foam with and without added flame retardant (FR) is well characterized and there is clearly a significant improvement. Although the synergy between SS and DMPY is modest, the overall study demonstrates a potential use for a major waste product, which lends merit to this study. There are a number of issues that should be addressed prior to publication of this work:

  • The authors use past tense throughout the manuscript when they should be using present tense. Whenever possible (and grammatically correct), present tense should be used to make the text more engaging. The first sentence of the abstract incorrectly uses past tense and there are many more (most) sentences throughout the manuscript that do this.

Reply:Thanks for your advice.

Manuscript has been revised, in which present tense is used.

  • In the second paragraph of the Introduction (on page 1), the sentence that begins with "Steel slag (SS) is rich..." should become the first sentence of this paragraph because it defines "SS". In the current organization, the abbreviation SS is used without being defined.

Reply:Thanks for your advice.

The abbreviation SS has been defined before use in Introduction part.

  • Section 2.3 can be one single paragraph.

Reply:Thanks for your advice.

The Section 2.3 has been revised into one single paragraph.

  • What is the average particle size of the SS used here? This should be measured and reported in the manuscript.

Reply:Thanks for your advice.

The average particle size of SS is 13 μm, which is reported in the manuscript.

  • The caption of Figure 1 should include a description of what the red circles and yellow arrows indicate.

Reply:Thanks for your advice.

The description about the red circles and yellow arrows has been added in the manuscript.

  • The references are not correct throughout the manuscript text. They appear as [0] in many locations.

Reply:Thanks for your advice.

The reference has been checked and revised throughout the manuscript.

  • The first sentence of Section 3.7 is not actually a sentence (i.e., it is an incomplete sentence). The entire manuscript text needs copyediting to fix clumsy English grammar.

Reply:Thanks for your advice.

The first sentence of Section 3.7 has been revised. And also, the English grammar has been enhanced throughout the manuscript.

  • In the Conclusions (on page 11), line 327 does not need to list the tools used for characterization. Deleting this list from the sentence will make it more concise and impactful. 

Reply:Thanks for your advice.

The tools used for characterization have been deleted in Conclusion part.

Reviewer 2 Report

Referee Report

on paper “Flame retarded rigid polyurethane foam composites with incorporation of steel slag/dimelamine pyrophosphate system: a new strategy for utilizing of metallurgical solid waste”

by Mingxin Zhu, Sujie Yang, Zhiying Liu, Shunlong Pan and Xiuyu Liu

Submitted to Molecules

This article is devoted to the study of flame retarded rigid polyurethane foam composites with incorporation of steel slag/dimelamine pyrophosphate system. The investigated composites were prepared using steel slag and dimelamine pyrophosphate as flame retardants. The results of combustion features investigation show a new idea for the high valuated utilization of composites and the improvement of flame retardant properties of Rigid polyurethane used as material for external wall insulation and cables due to its good thermal insulation performance.  In general, the studies were carried out at a high technical level, the article was well constructed and easy to read. However, a sufficiently in-depth analysis was not carried out. This may cast doubt on the conclusions. Given the environmental focus and interesting results, the article may be of great interest to readers. Thus, my decision is major revision.

Comments

1.                  First of all, English should be improved.

2.                  The introduction is very poor and contains only general information about rigid polyurethane, dimelamine pyrophosphate and composites based on it. Only 13 publications are considered in the introduction. The introduction should be expanded by considering more papers on the topic.

3.                  In the article, there is often a reference to the source at number 0, if I understand everything correctly and this has no other purpose. It needs to be corrected

4.                  I ask the authors to explain why the strength varies non-linearly depending on the composition of the composites. I believe that this is tightly connected with the structure, the analysis of which also needs to be done in more depth.

5.                  The results of the study of morphology by the SEM method show significant changes in the structure of the composites. One of the important parameters for composite and many other materials is porosity. In this study, the authors do not pay attention to this, however, it determines not only the main functional properties, but also operational and mechanical characteristics, material density, and much more. An analysis of the trend in porosity will help to explain the change in some other characteristics and make the conclusions more reasonable. I recommend that the authors read some of the papers related to this and use the methodology described in them to determine the porosity.  doi:10.3390/nano10061245, https://doi.org/10.3390/nano12121998, https://doi.org/10.1021/acsomega.1c00611

6.                  The TG-FTIR 3D images of RPUF and RPUF-R composites in Figure 5 look like the experimental scan error. Please explain why the z-axis deviations are so large and statistically dispersed. I also believe that filtering along the time axis should be done for better visualization.

7.                  The resolution of images in the figures 5-6 needs to be improved

Author Response

Journal: Molecules

Manuscript Number: molecules-2017594

Title: Flame retarded rigid polyurethane foam composites with incorporation of steel slag/dimelamine pyrophosphate system: a new strategy for utilizing of metallurgical solid waste

Author(s): Mingxin Zhu, Sujie Yang, Zhiying Liu, Shunlong Pan, Xiuyu Liu

Thanks a lot for your attention and the reviewers’ evaluation and comments on our manuscript (molecules-2017594). We have modified the manuscript according to your helpful advice and reviewers’ detailed suggestions with the correction sections. At the same time, the grammatical errors of this manuscript have been carefully revised. Please find the point-by-point response to the reviewers.

Yours sincerely,

Mingxin Zhu

Reviewer 2

This article is devoted to the study of flame retarded rigid polyurethane foam composites with incorporation of steel slag/dimelamine pyrophosphate system. The investigated composites were prepared using steel slag and dimelamine pyrophosphate as flame retardants. The results of combustion features investigation show a new idea for the high valuated utilization of composites and the improvement of flame retardant properties of Rigid polyurethane used as material for external wall insulation and cables due to its good thermal insulation performance.  In general, the studies were carried out at a high technical level, the article was well constructed and easy to read. However, a sufficiently in-depth analysis was not carried out. This may cast doubt on the conclusions. Given the environmental focus and interesting results, the article may be of great interest to readers. Thus, my decision is major revision.

  • First of all, English should be improved.

Reply:Thanks for your advice.

English has been enhanced throughout the manuscript.

  • The introduction is very poor and contains only general information about rigid polyurethane, dimelamine pyrophosphate and composites based on it. Only 13 publications are considered in the introduction. The introduction should be expanded by considering more papers on the topic.

Reply:Thanks for your advice.

The Introduction part has been enhance in the manuscript.

  • Inthe article, there is often a reference to the source at number 0, if I understand everything correctly and this has no other purpose. It needs to be corrected

Reply:Thanks for your advice.

It has been revised in the manuscript.

  • I ask the authors to explain why the strength varies non-linearly depending on the composition of the composites. I believe that this is tightly connected with the structure, the analysis of which also needs to be done in more depth.

Reply:Thanks for your advice.

We are very consistent with the reviewer. There may be some misrepresentation in the manuscript, the strength is affected by many factors, including the composition, structure. Especially to the structure of polyurethane foam composites with many cell structures, flame retardant particles, the foaming process, cell structures will result in complication of the strength variation. The further investigation is conducting. However, because of Covid 19, the results are on the way.

  • The results of the study of morphology by the SEM method show significant changes in the structure of the composites. One of the important parameters for composite and many other materials is porosity. In this study, the authors do not pay attention to this, however, it determines not only the main functional properties, but also operational and mechanical characteristics, material density, and much more. An analysis of the trend in porosity will help to explain the change in some other characteristics and make the conclusions more reasonable. I recommend that the authors read some of the papers related to this and use the methodology described in them to determine the porosity.doi:10.3390/nano10061245, https://doi.org/10.3390/nano12121998, https://doi.org/10.1021/acsomega.1c00611

Reply:Thanks for your advice.

The porosity is very important to cellular materials, which could significantly affect the mechanical properties, density, thermal conductivity, combustion behavior of the composites. Herein, we are trying to investigate the porosity of the RPUF composites according to the above research. However, because of the Covid 19, the corresponding data are on the way.

The related reference are cited and discussed in the manuscript.

  • The TG-FTIR 3D images of RPUF and RPUF-R composites in Figure 5 look like the experimental scan error. Please explain why the z-axis deviations are so large and statistically dispersed. I also believe that filtering along the time axis should be done for better visualization.

Reply:Thanks for your advice.

There may come from the following reasons: 1)The accidental volatile components in the decomposition of the composites; 2) the deteroriation of working conditions of TG-FTIR.

The filtering along the time axis has been conducted,but the effect is not obvious.

Absorbance of pyrolysis products of RPUF and FR-RPUF versus time: a) Gram-Schmidt; b) CO2; c) isocyanate compound;d carnonyl compound; e) aromatic compound; f) esters

Thus, we conduct the filtering along the wavenumber axis, which is added in the manuscript.

  • The resolution of images in the figures 5-6 needs to be improved.

Reply:Thanks for your advice.

The resolution of images in the Figures 5-6 has be improved.

Round 2

Reviewer 2 Report

Accept as is